# Where to Charge Electric Trucks in Europe—Modelling a Charging Infrastructure Network

**Daniel Speth** * , **Verena Sauter and Patrick Plötz**

Fraunhofer Institute for Systems and Innovation Research ISI, Breslauer Strasse 48, 76139 Karlsruhe, Germany
* Correspondence: daniel.speth@isi.fraunhofer.de; Tel.: +49-721-6809-243

**Abstract:** Heavy-duty trucks account for 27% of the European greenhouse gas emissions in the transport sector. To decarbonize road freight transport, the European Union plans to build a fast charging network for trucks. This paper presents two scenarios, covering European highways with charging stations at regular intervals every 50 or 100 km along the most important highways. For each location, the required number of charging points at 15% battery electric trucking is calculated individually using queueing theory. A third scenario takes into account the infrastructure ramp-up in 2025 and assumes a share of 5% battery electric trucking in a network with a 100 km distance. We define a network of 660 (100 km distance) or 1468 stations (50 km distance). Depending on the scenario and the individual station, the projected number of charging points per station varies between 1 and 18 in 2030. The results give a first insight into what a fast charging infrastructure for trucks in Europe might look like. In particular, we show that large charging stations with more than ten charging points could be necessary in the next few years. This knowledge might help to design future charging infrastructure for electric road freight transport.

**Keywords:** battery electric truck; charging infrastructure; heavy-duty trucks; infrastructure; network modelling; queueing theory

## 1. Introduction

Road transport causes approximately one quarter of current European greenhouse gas (GHG) emissions. In turn, heavy-duty trucks and buses account for 27% of road transport GHG emissions [1]. European Union (EU) legislation requires $CO_2$ emissions from newly registered heavy-duty vehicles to be reduced by 30% in 2030 compared to current levels [2]. Analyses show that this is only possible when using zero emission vehicles (ZEV), i.e., electric or hydrogen powered trucks [3]. When comparing different options to electrify heavy-duty trucks (e.g., fast charging, overhead catenary or battery swap), fast charging seems to be the best option in the short term, mainly due to a comparatively scalable infrastructure [4]. Current research shows that charging infrastructure is essential for the diffusion of battery electric trucks (BET) [5,6]. Consequently, the EU plans to install a fast charging infrastructure for BET [7].

According to [8,9], charging infrastructure planning models can be divided into three groups: node-based, path-based, and tour-based models.

Prominent representatives for node-based models are p-median models, where p facilities are positioned at nodes so that the demand of neighboring nodes can be fulfilled with a minimum distance travelled [9]. The approach is based on [10], who used it to determine the optimal position of police stations in a road network. A further development is the set-covering problem, which ensures that every demand can be served. Ref. [11] originally used this approach to position emergency service facilities. This approach can also be used for the positioning of charging locations for electric vehicles [12]. In node-based models, a charging station covers a certain area or a certain part of a road. Therefore, typically a quite dense charging network with many charging points is modelled.

The coverage approach described in [13] represents a special case. Unlike the previous approaches, charging locations are placed at regular intervals and sized based on the traffic volume determined for the neighboring nodes. Therefore, the focus is not on minimizing the number of charging locations but on sizing them. Ref. [14] shows a coverage approach to design an infrastructure for battery electric cars in Germany. Ref. [13] transferred the approach to BET in Germany. Like all node-based models, the coverage approach benefits from low data requirements [9]. Only the traffic volume at the nodes is required [13].

In contrast, path-based models rely on traffic flows within a network and try to cover a maximum of passing traffic with a minimum of stations. The Flow Capturing Location Model (FCLM), the first subgroup of path-based models, was introduced by [15]. For example, [16] using a FCLM to position 27 charging stations for battery electric cars in the city of Barcelona would serve 92% of the considered flows. Ref. [17] introduced the Flow Refueling Location Model (FRLM) that also considers multiple stops for one path. The general idea of placing locations such that a maximum of origin–destination paths known in advance can be supplied with a fixed number of stations remains the same. FRLM has been used to model charging infrastructure for battery electric cars in the USA [18] or in Europe [19]. Ref. [20] transferred this approach to hydrogen powered trucks and combined the model with a capacity restriction to avoid unrealistic large stations. However, due to the high computational effort, restrictions are usually necessary. For example, [18] clustered 4486 regions to 196 regions. Ref. [19] reduced the problem by ignoring flows with less than 5000 vehicles per year.

Finally, a tour-based model considers individual driving profiles and locates charging stations such that they fit the driving profiles. While the level of detail increases from the node-based to the path-based to the tour-based models, the demands on the input data also increase. For the node-based approach, data from local traffic counts are sufficient. Path-based models require origin–destination relations. The tour-based models typically rely on journey logs. For example, [21] using driving trip data to model slow-charging infrastructure for cars in the city of Columbus (OH, USA). For a deeper comparison of the models, refer to [8,9].

Regarding infrastructure modelling for trucks, initial publications already exist. Ref. [22] used GPS data from eight million vehicle trips and implemented a path-based model with individual trip data for short-haul BET in South East Queensland. The vehicle trips were clustered to 13,456 origin–destination paths and 116 possible charging locations were identified. Using up to 10 optimally positioned realized charging locations, 85 charging networks were modeled and a network coverage of up to 93% was derived. Also using a path-based model, [20] modelled a hydrogen refueling infrastructure for trucks in Germany. The analysis was based on 2655 origin–destination paths. As a result, 100 refueling stations can serve Germany (13,000 km highway) without any capacity restrictions. Depending on the number of hydrogen trucks, a capacity restriction, e.g., a maximum daily capacity per refueling station, can significantly increase the number of refueling stations. In contrast, [13] extended the coverage approach for battery electric cars from [14] and designed a public fast charging infrastructure for BET in Germany. Depending on the specified distance, 142 to 267 charging stations were identified. At 15% BET trucking, these charging stations are equipped with 2 to 13 charging points per station.

As shown, previous analyses usually consider geographically small areas and/or are based on significant simplifications. To the best of the authors' knowledge, there is no publication that presents a European fast charging infrastructure for BET. Since the EU is currently debating a Europe-wide charging network for trucks [7], it is highly important to develop an idea of what such a network could look like in the next few years.

Therefore, the aim of this paper is to design a public fast charging network for BET throughout Europe for the mid-term future (e.g., 2030) based on transport flows. We limit the analysis to the EU, extending to Great Britain, Switzerland and Norway (EU27+3). Since the proposal for the Alternative Fuel Infrastructure Regulation (AFIR) [7] suggests charging infrastructure at regular intervals along the European highway network, we follow [13]

and apply a node-based approach and combine it with a queuing model. The queueing model determines the number of necessary charging points for a given share of BET for each charging station.

This paper differs from existing studies in several aspects. First, the geographical coverage includes the entire EU as well as Great Britain, Switzerland and Norway. Publicly available data is used for this purpose. Second, the paper follows the logic of the AFIR proposal [7] and models charging infrastructure at regular intervals. Third, the approach extends beyond location identification and focuses on the sizing of the locations using queueing theory. Fourth, the scenarios in this paper show various options and their implications for a potential short- to medium-term public infrastructure for trucks.

The paper is structured as follows: First, we present relevant input data and the methodological approach. Afterwards, Section 3 contains the results. Finally, we discuss some critical aspects in Section 4 and conclude our work in Section 5.

## 2. Materials and Methods

### 2.1. Materials and Scenarios

To model a charging infrastructure for Europe, we require the traffic volumes on the European highway network. Since homogeneous traffic counts for all European countries are rarely available and the standardization of country-specific data would be very burdensome, modelled European traffic flows [23] serve as our basis. The dataset represents an update of the truck traffic flows published in the European Transport policy Information System (ETIS) project [24] and contains projections between NUTS3 (Nomenclature des unités territoriales statistiques. Level 3: small regions, large cities)-regions in Europe. The dataset contains only transports between NUTS3-regions. Inner regional traffic is not included. However, inner regional traffic is not relevant for public fast charging infrastructure on the highway network. In the following, we refer to the truck traffic flows in the dataset as long-haul traffic. The underlying road network contains 17,435 nodes that are connected with 18,447 edges. In some sections, we add additional nodes to ensure a maximum distance of 10 km between two nodes. The nodes serve—simplified—as potential station locations in this paper. For each edge and node, the number of vehicles that pass them per year in 2010, 2019 and 2030 is known. In the following, we refer to those edges as subsections of one road. Figure 1 shows the traffic volume in Europe in 2030 according to [23]. Within this paper, we filter the updated ETIS dataset and focus on the international E-road network in Europe.

Additionally, we derive a cumulative annual mileage of 162,397 million km in 2019 and 215,042 million km in 2030 for long-haul traffic on roads in the EU, including Switzerland, Norway and Great Britain from [23]. Simplified, we assume 188,719 million km in 2025.

As shown in [13], and based on the automated traffic census in Germany [25], we assume that a maximum of 6% of the daily charging events happen in the most trafficked hour of the day. For this hour, the average waiting time should not exceed 5 min, according to experts from the automotive industry [26]. Thus, the charging process of 30 min can be carried out within the mandatory break of 45 min after 4.5 h of driving, including a 10 min buffer. Therefore, a recharge for approximately 300 km is required each time [13].

We assume that 25% of the charging events occur on public fast charging infrastructure. Survey data [27] show that about half of the heavy-duty vehicles drive less than 500 km per day [4]. For these vehicles, we assume that they are almost exclusively charged at the depot. For the other vehicles, we assume that half of their charging events take place at the origin or destination depot. This means that, on average, half of the BET use public infrastructure and, for these trucks, every second charging process takes place publicly. However, this assumption comes with a high level of uncertainty. As shown in [13], the required number of charging points increases approximately linearly with the share of public charging events. Table 1 sums up the most important parameters.

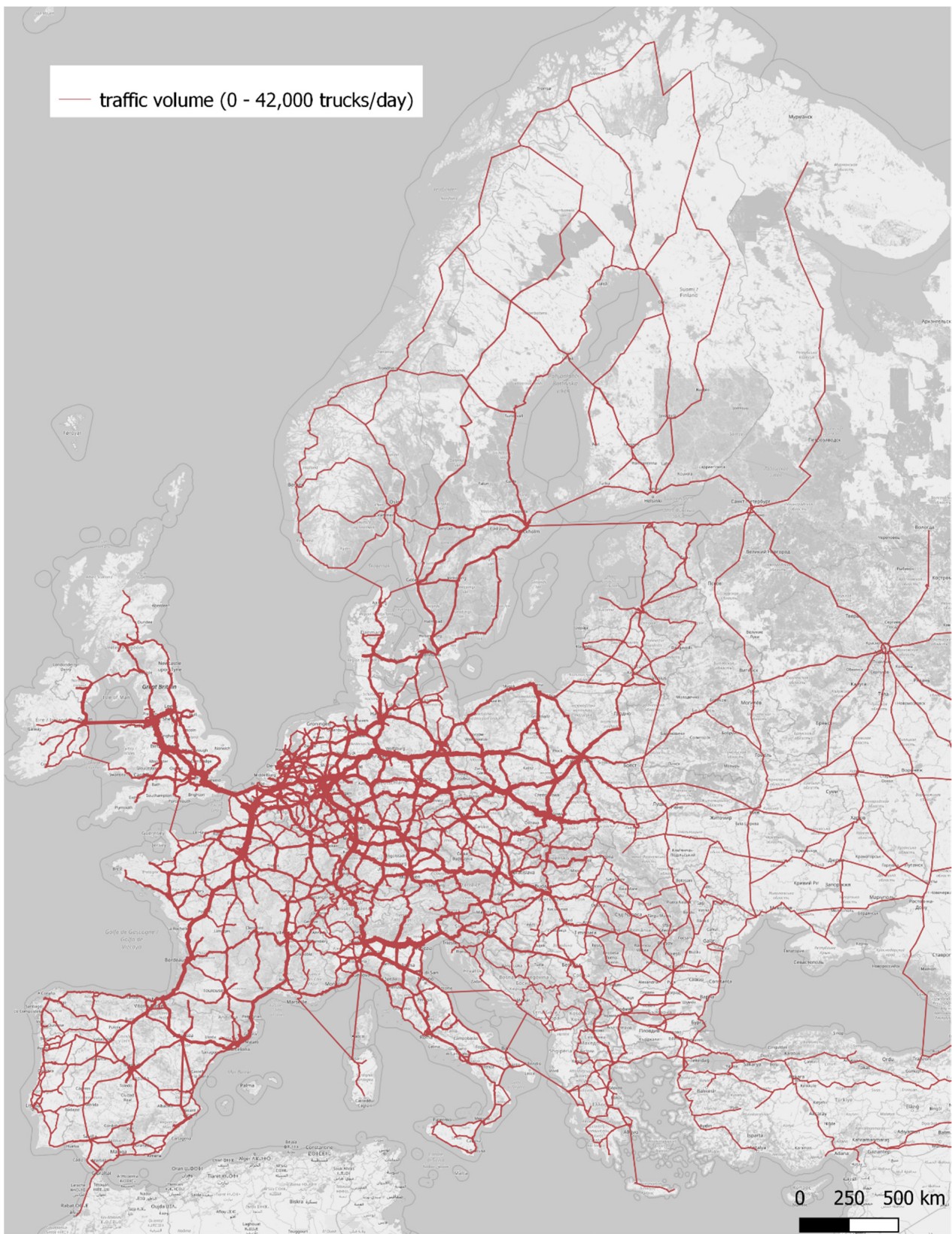

**Figure 1.** Traffic volume in Europe 2030. Own illustration based on [23]. Background: OpenStreetMap.

**Table 1.** Input parameters for infrastructure calculation.

| Parameter | Abbreviation | Value | Reference |
|---|---|---|---|
| Cumulative annual mileage | $AM_{HDV,EU27+3}$ | 188,719 Mio. km (2025)<br>215,042 Mio. km (2030) | Own calculation, based on [23] |
| Range in 4.5 h | $range_{BET}$ | 300 km | [13] |
| Share of public charging | $CE_{public}$ | 25% | Own estimation, based on [13,26] |
| Average charging time | | 30 min | [13] |
| Average waiting time | $W_q$ | 5 min | [13] |
| Share of daily charging events in most trafficked hour | | 6% | [13] |

In the following, we will focus on three scenarios: First, we will design a startup network for 2025. Within the startup network, we assume 5% of the annual mileage being electrified ($BET_{share} = 0.05$), following [3] and expert opinion [26]. Additionally, we assume a distance of $d_{avg} = 100$ km between the charging locations. This is slightly more than the EU's proposal with 60 km [7]. Second, we will design an expansion network for 2030 that will densify the startup network to 50 km, by putting additional stations between the stations of the startup network. The $BET_{share}$ will grow to 15%. Third, we will design a widemeshed network for 2030 with a distance of 100 km and a $BET_{share}$ of 15%. Table 2 sums up the most important information. Figure 2b illustrates the relationships between the three scenarios.

**Table 2.** Scenario definition.

| Scenario | Targeted Year | $d_{avg}$ | $BET_{share}$ |
|---|---|---|---|
| Startup | 2025 | 100 km | 5% |
| Expansion | 2030 | 50 km | 15% |
| Widemeshed | 2030 | 100 km | 15% |

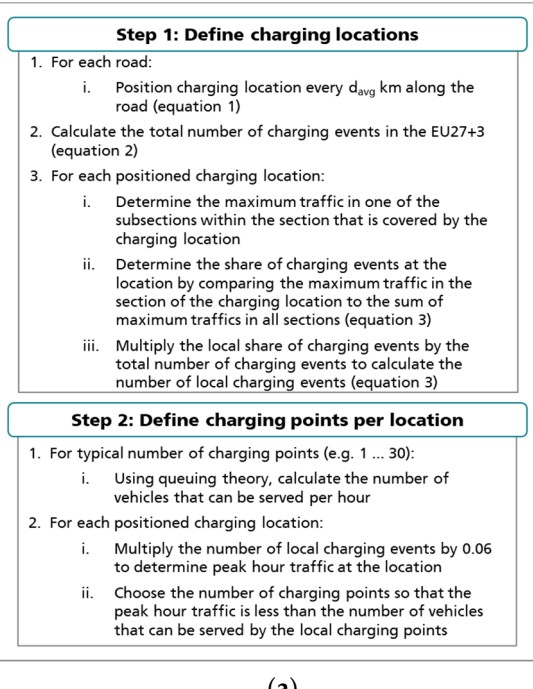

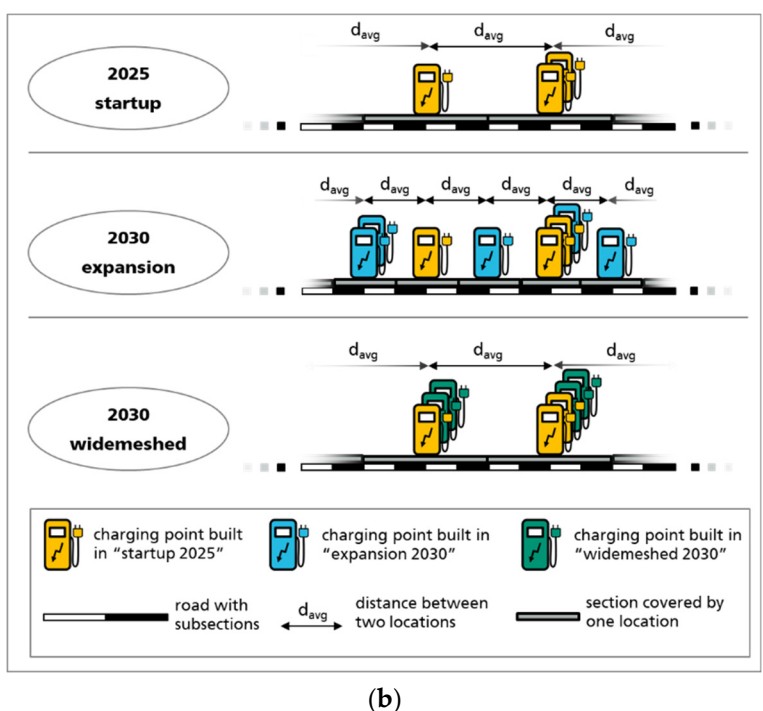

(**a**)     (**b**)

**Figure 2.** Methodological procedure (**a**) and simplified draft (**b**) for the calculation of the charging networks.

### 2.2. Methods

The methodological procedure is divided into two steps: First, the charging locations and the number of charging events at every location are determined. Second, the number of charging points for every location is calculated. An overview of the methodological procedure is given in Figure 2a. The steps are described in more detail in Sections 2.2.1 and 2.2.2. For more information, compare [13].

### 2.2.1. Determine Charging Locations

According to the coverage approach described in [13], every single road in the E-road network graph is traversed successively according to a predefined scheme. The road network from [23] serves as our basis. Within the E-road network, there are two major groups: odd numbers indicate roads that run from north to south, whereas even road numbers run from east to west. Each of these roads is processed one after the other in ascending order, along the previously defined direction of travel. Every node in the network is a potential charging location. Locations are positioned at regular intervals. Equation (1) shows this approach, where $CL_L$ is a bivariate variable indicating whether infrastructure is built in $L$ or not. $d_{CL,L}$ indicates the distance between the last positioned charging location and location $L$. $d_{avg}$ defines the distance between two charging stations in the network.

$$CL_L = \begin{cases} 1, \ if \ d_{CL,L} \geq d_{avg} \\ \quad 0, \ else \end{cases} \tag{1}$$

Afterwards, the total number of daily public charging events in the EU27+3 $CE_{EU27+3}$ is calculated as follows:

$$CE_{EU27+3} = \frac{BET_{share} * (AM_{HDV,EU27+3}/313)}{range_{BET}} * CE_{public} \tag{2}$$

$BET_{share}$ stands for the share of BET on the total cumulative annual mileage $AM_{HDV,EU27+3}$ of all heavy-duty trucks. The annual mileage is divided by 313 to derive daily mileage, excluding Sundays. $range_{BET}$ refers to the range that a truck can cover in 4.5 h of driving. This corresponds to the maximum driving time before a mandatory break is required. Finally, we multiply this with the share of charging events on public infrastructure $CE_{public}$.

Finally, the expected daily public charging events have to be allocated to individual charging locations. For this purpose, the maximum traffic volume in the area in front of and behind the location is calculated and compared with the total maximum traffic volume of all locations. The number of trucks in both directions is considered together. The 04_network_edges dataset described in [23] serves as the basis for the subsection-by-subsection traffic volume. Equation (3) describes the calculation of the daily charging events at each realized charging locations. $MAX_{CL_{i-0.5}}^{CL_{i+0.5}}(TV_j)$ describes the maximum traffic volume of all of subsection $j$ on half the distance between the realized charging location $i$ and the realized station before this location ($CL_{i-0.5}$) and half the distance to the subsequent location ($CL_{i+0.5}$). The individual maximum traffic volume is set in relation to the sum of all maximum traffic volumes of all realized stations.

$$CE_{CL_i} = CE_{EU27+3} * \frac{MAX_{CL_{i-0.5}}^{CL_{i+0.5}}(TV_j)}{\sum_{CL} MAX_{CL_{i-0.5}}^{CL_{i+0.5}}(TV_j)} \tag{3}$$

### 2.2.2. Dimension Charging Locations

The calculation of the number of charging points per location is based on queueing theory. For the peak hour, we assume 6% of the daily charging events of one location, as described above. The system is designed for this size. We stick to the Kendall notation ($A/S/c/d/k/m$), to define the queueing system. To define the arrival process $A$, we assume Poisson-distributed arrivals [28], with the average arrival rate $\lambda = CE_{CL_i} * 6\%$. The inter-arrival times are therefore exponentially distributed (Markovian Distribution M). This means $A$ = M. With regard to the service process $S$, [28] show that a General distribution G with normally distributed service times fits quite well. The average number of customers served per period is defined by $\mu$. For example, an average charging time of 30 min results in an average service rate $\mu$ = 2 trucks/hour. The number of service units $c$—charging points—shall be calculated. For $D$, $k$, and $m$, we assume the default values. The queue's discipline $d$ follows the First-In-First-Out principle. This means that the trucks are served in the order of their arrival. The number of customers waiting in the queue $k$ is assumed to be infinite. The same applies to the number of customers in total. Therefore, we define an M/G/c queueing system. Since the exact solution for the mean waiting time of M/G/c systems is not known, the mean waiting time is approximated, according to [29]:

$$W_q^{M|G|c} = \frac{C^2 + 1}{2} W_q^{M|M|c} \tag{4}$$

$C$ is defined as the variation coefficient of the distribution of the service times, i.e., the standard deviation (5 min) divided by the mean value of the service time distribution (30 min). This formula is used with the waiting time of the original M/M/c system, given in Equation (5):

$$W_q^{M|M|c} = \frac{1}{1-\rho} \frac{1}{c\mu} \frac{(c\,\rho)^c}{c!} \left( (1-\rho) \sum_{n=0}^{c-1} \frac{(c\rho)^n}{n!} + \frac{(c\rho)^c}{c!} \right)^{-1} with\ p = \frac{\lambda}{c\mu} \tag{5}$$

Finally, we calculate the maximum average arrival rate $\lambda$ that allows for an average waiting time of 5 min for all possible numbers of charging points $c$. For each location, we compare the local average arrival rate $\lambda = CE_{CL_i} * 6\%$ to the arrival rates with different $c$. Afterwards, we can choose the number of charging points $c$ for each location so that the average waiting time is less than 5 min.

## 3. Results

Figures 3 and 4 show the distribution of the charging locations in the EU27+3. For the startup network and the widemeshed network with a distance of 100 km, we receive 660 charging locations. For the expansion network with a 50 km distance, we count 1486 charging locations. There is more than a doubling since very short sections as well as peripheral areas also receive charging locations in the closer network.

With regard to the geographical distribution of the charging points, there is a concentration on Central Europe (e.g., France and Germany). The traffic generated by the ports in the Netherlands, Belgium and Germany is of great interest for the dimensioning of charging locations. The surrounding countries (e.g., Norway, Sweden, Finland, Greece, Italy and Spain) are equipped with smaller locations that cover the whole area.

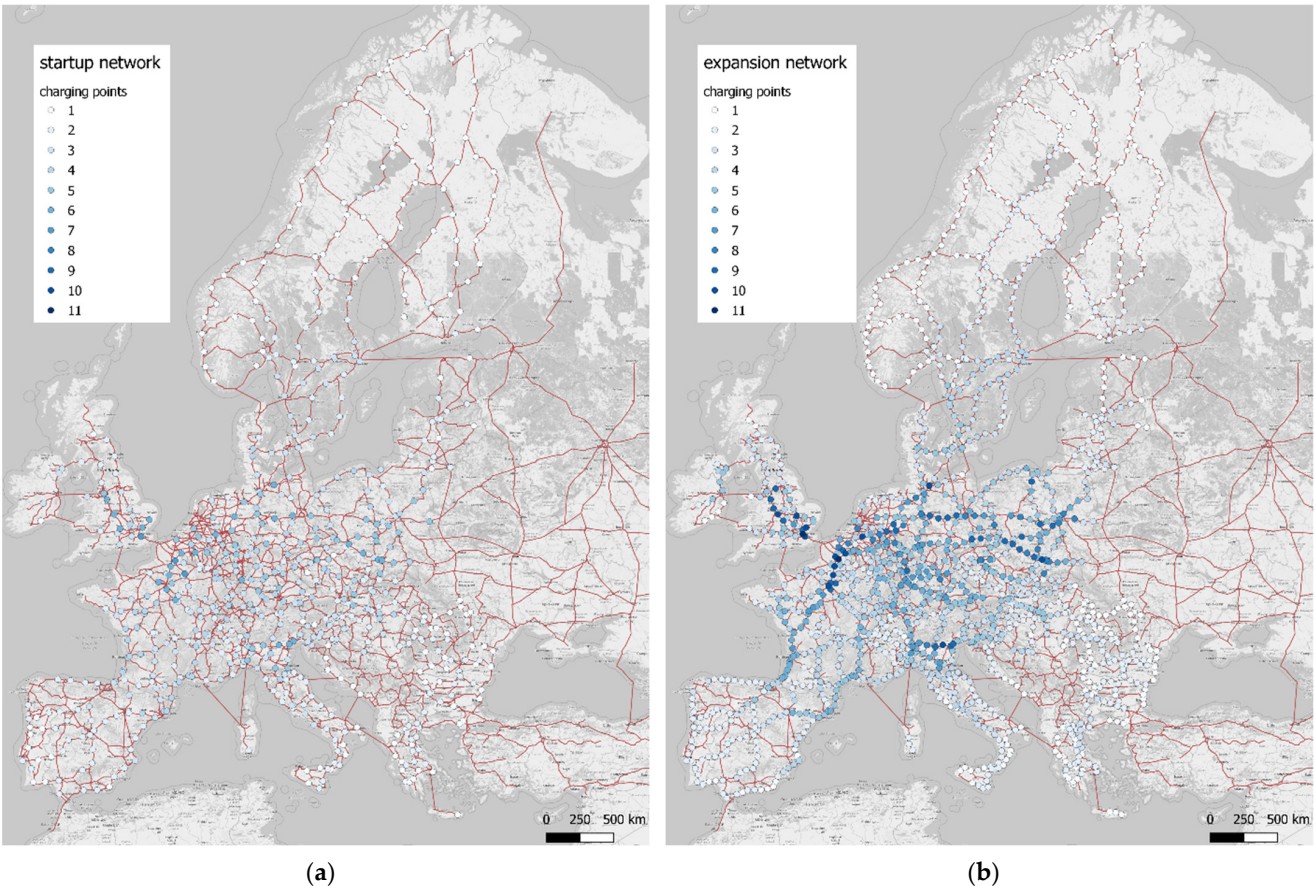

**Figure 3.** Location of 660 charging locations in the startup network (**a**) and 1468 charging locations in the expansion network (**b**). Background: OpenStreetMap.

The startup network consists of 1697 charging points at 660 locations. This means that an average charging location includes 2–3 charging points in 2025 (mean = 2.57, median = 2). Assuming a tripling of electric traffic ($BET_{share}$ 5% versus 15%) and a densification of the network from 100 km to 50 km, the average charging location still has three charging points (mean = 3.25, median = 3). However, the largest station contains 11 charging points instead of seven. We refer to this scenario as expansion network. In total, 4778 charging points are needed in this scenario. Figure 5a shows the change in the required number of charging points from the startup network to the expansion network. In less than 2% of all charging locations, between two and four charging points are removed. These are typically locations nearby areas with high traffic, such as ports, where a new charging location is opened closer to the high traffic location. If the network is not densified, 3679 charging points are required, as shown in the widemeshed network. However, the individual charging locations will be significantly larger. An average charging location is equipped with 5–6 charging points (mean = 5.57, median = 5). The biggest charging location consists of 18 charging points. Figure 5b shows the distribution of charging points among the individual charging locations for all scenarios.

Since our model relies on assumptions, Figure 6 shows a sensitivity analysis for $BET_{share}$, $CE_{public}$, and $range_{BET}$. We varied the parameters by +/−50%. An increase in $BET_{share}$ or $CE_{public}$ by 50% increases the number of charging points from 4778 to 6211. This means the number of charging points increases by 30%. The same effect can be observed when reducing $BET_{share}$ or $CE_{public}$. A reduction of $range_{BET}$ by 50% leads to an increase of charging points from 4778 to 7.581. An increase of $range_{BET}$ by 50% reduces the number of charging points to 3813.

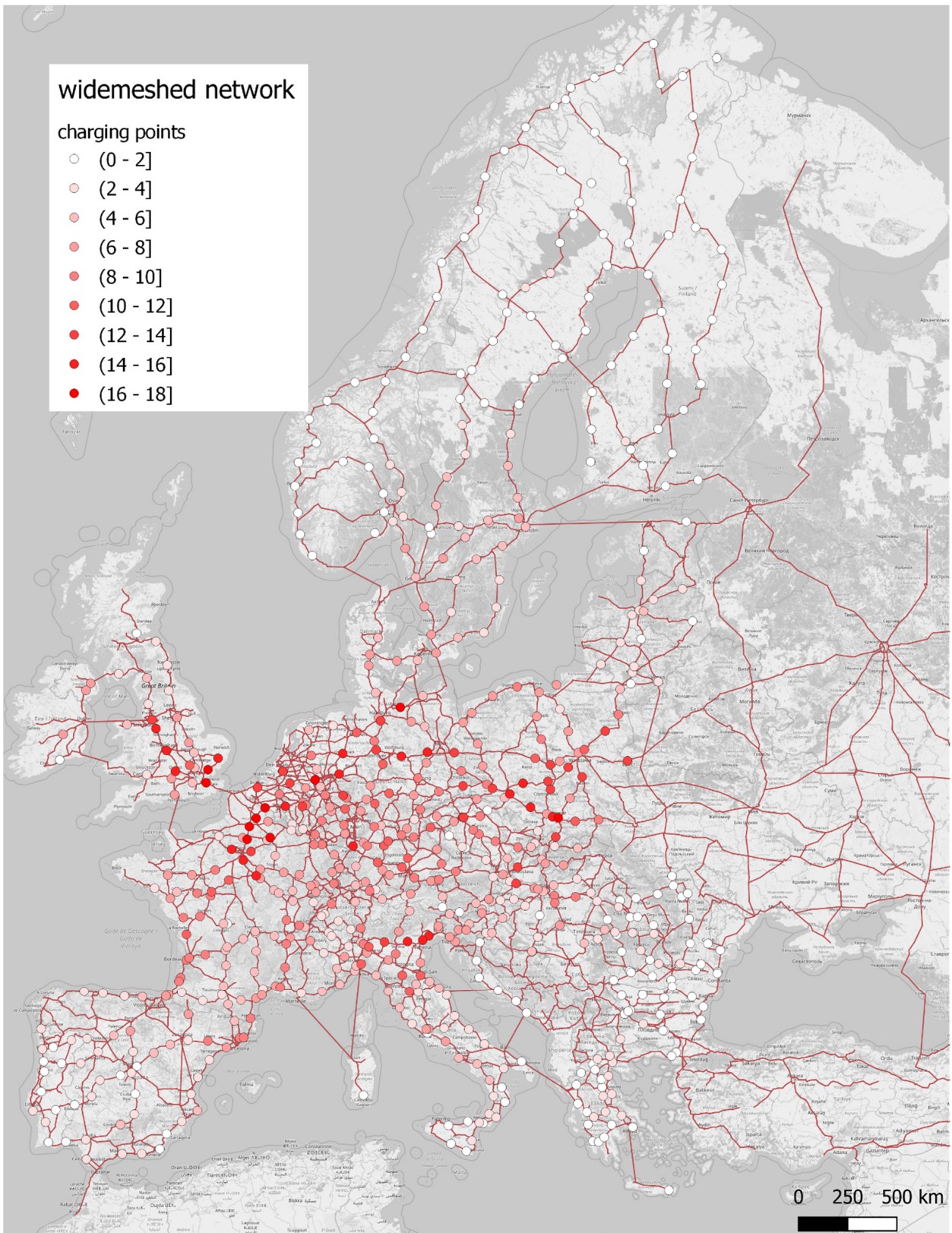

**Figure 4.** Location of 660 charging locations in the widemeshed network. Background: OpenStreetMap.

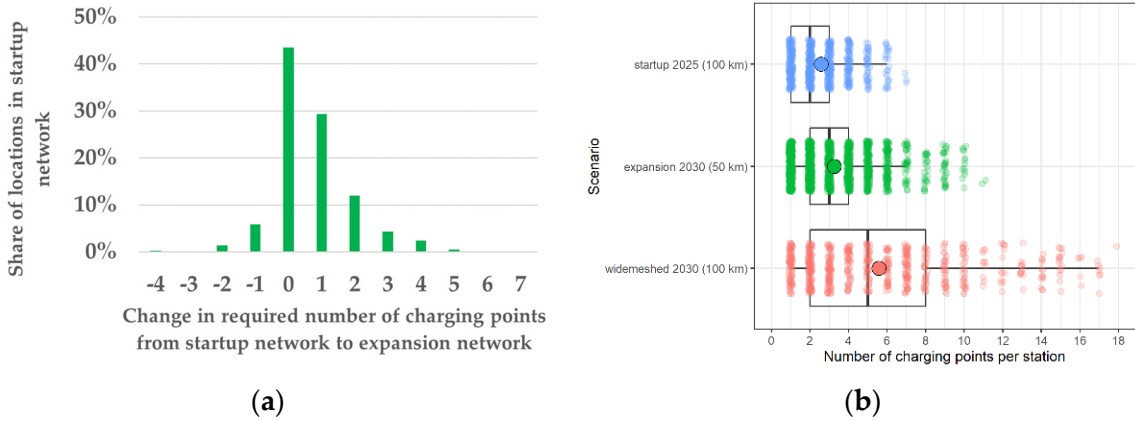

**Figure 5.** (**a**) Change in number of charging points from startup network to expansion network. (**b**) Boxplot of charging points per location for all scenarios.

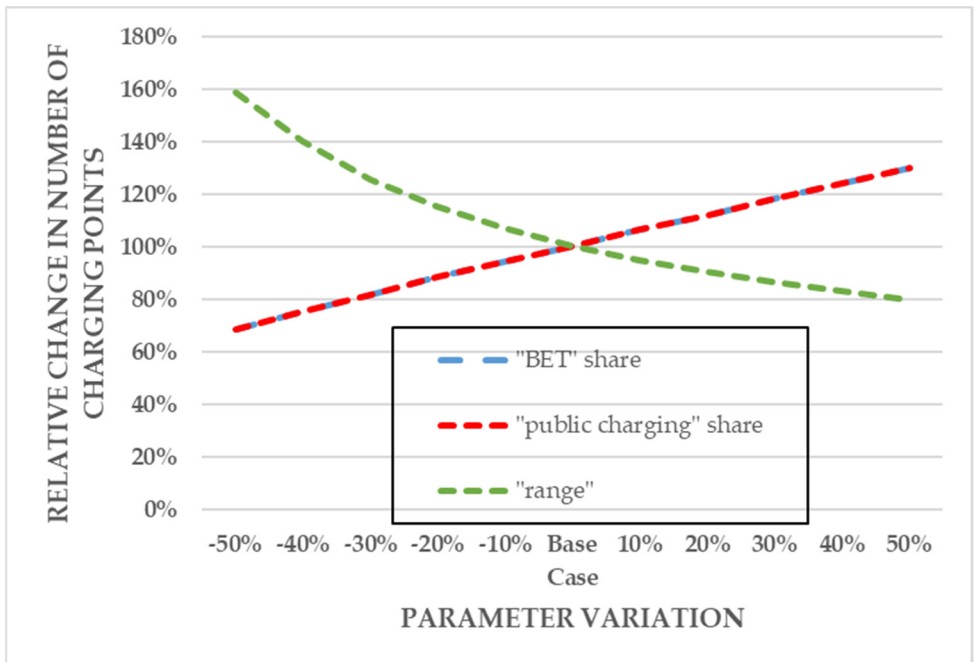

**Figure 6.** Sensitivity analysis for the expansion network.

## 4. Discussion

### 4.1. Data Limitations

The results shown in this paper rely on different assumptions and data sources. In particular, assumptions regarding the share of battery electric trucking and the vehicle range are currently based on expert estimates. As shown in Figure 6, the number of charging points relies on these assumptions. The effects of under or overestimation are remarkable but are not of a different order of magnitude. Therefore, the assumptions should be further investigated and validated in the future. Coupling the infrastructure ramp-up with a market diffusion model for trucks could also improve the reliability compared to our predefined market diffusion scenarios.

In contrast to previous work [13], the traffic volumes in this paper are based on synthetic road freight transport flow data [23]. On the one hand, this significantly improves the results, since regional traffic that is unlikely to use public fast charging infrastructure is not included in the dataset. On the other hand, the dataset itself relies on significant simplifications, e.g., in terms of resolution, scaling, and non-consideration of multi-stop-tours. From a scientific perspective, additional data sources, such as driver logbooks,

should be integrated to better reflect actual driving behavior. From a planner's perspective, activities at local rest areas should be observed to validate the model results.

While [13] uses the entire German highway network as potential locations for charging stations, we restrict our analysis to the E-road network. This reduces the number of charging stations from 267 [13] to 154 in the 50-km network (expansion network) and from 142 [13] to 67 in the 100-km network (widemeshed network) in Germany. Given the same assumptions in both models, the number of charging points in total remains almost identical for Germany. This shows the quality of the synthetic dataset [23] compared to the real traffic count data used in [13]. As in the comparison between the expansion network and the widemeshed network, it also becomes clear that the average number of charging points per station depends on the underlying network distance and density.

*4.2. Model Limitations*

In addition to the data limitations, the model itself also comes with some limitations. First, the location selection does not take into account the suitability of the location for a charging area. Aspects such as parking area availability or the power grid connection are not part of the analysis. The locations are intended as representatives for the particular highway section, not as a defined location. However, the model gives a good impression of the general distribution of charging locations as well as the total number of charging points required. As part of the model development, location details could be integrated up to a certain level in the future. However, a planner will evaluate the local conditions in the targeted area in detail.

Second, as a node-based model, the coverage approach does not consider traffic flows. The origin–destination paths contained in the synthetic dataset are therefore converted into traffic volumes per subsection of a road. Vehicles are counted several times for different subsections they pass. It is plausible that vehicles recharge evenly distributed throughout the road network according to the local traffic volume. However, special effects may occur. As an example, the underlying dataset contains extensive port-hinterland-traffic. This traffic leads to a high traffic volume locally. As shuttle transports, these vehicles are probably charged at private depots. The model tends to overestimate the required public charging infrastructure in these sections. The assumption that charging processes are distributed equally to traffic volumes should be verified in the future, for example with driving logbooks or with data from parked vehicles. To overcome the underlying problem, the use of path- or tour-based models can also improve the results if the computation time allows it without too much simplification.

Third, the differentiation between public fast charging in the mandatory break and public slow charging is highly simplified and can only be modeled as part of $CE_{public}$. At this point, path-based or tour-based models are better suited. However, these models are associated with significantly higher computational effort and a high demand on data availability.

**5. Conclusions**

In conclusion, a dense fast-charging infrastructure for heavy-duty trucks is essential for the successful electrification of road freight transport. The EU is currently defining the legal framework to create a corresponding network. This paper gives a first insight into what a fast-charging network in Europe with charging stations at regular intervals might look like. Our results indicate a demand of approximately 700 to 1500 charging locations, with up to 4800 charging points in total within the next few years. In addition, the paper shows that even in an early stage of market, large locations with more than ten charging points are required. The size of the charging stations depends on the distance between the individual stations. In the first years of battery electric trucking, a dense network—e.g., the expansion network with 50 km distance between the charging stations—can avoid large stations with significantly more than ten charging points. Given the EU's targets for the

$CO_2$ reduction of newly sold vehicles in 2030, technical issues should be clarified, path decisions should be made and construction projects should be initiated.

From a scientific perspective, we have shown a first approach to estimate the required charging infrastructure even for large areas. Future research should focus on using path- or tour-based models to analyze the future needs for public truck charging infrastructure. This would provide an even more detailed view on how to locate and dimension individual charging stations.

**Author Contributions:** Conceptualization, D.S. and P.P.; methodology, D.S., V.S. and P.P.; software, D.S. and V.S.; validation, D.S. and P.P.; formal analysis, D.S.; investigation, D.S.; data curation, D.S. and V.S.; writing—original draft preparation, D.S.; writing—review and editing, P.P.; visualization, D.S.; supervision, P.P.; project administration, P.P.; funding acquisition, P.P. All authors have read and agreed to the published version of the manuscript.

**Funding:** The works in this paper were supported by EU STORM project funded from the European Union's Horizon 2020 research and innovation program under grant agreement No 101006700. The Federal Ministry of Transport and Digital Infrastructure (BMVI) Germany funded this research within the project HOLA (FKZ 03EMF0404A).

**Data Availability Statement:** Not applicable.

**Conflicts of Interest:** The authors declare no conflict of interest. The funders had no role in the design of the study; in the collection, analyses, or interpretation of data; in the writing of the manuscript; or in the decision to publish the results.

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
