# Peer review of "Where to Charge Electric Trucks in Europe—Modelling a Charging Infrastructure Network"

_wevj, doi:10.3390/wevj13090162_

Round 1

Reviewer 1 Report

The submitted paper is well-presented. The detailed comments are listed as follows:

1. More literature review is necessary.

2. A diagram of the determination method is recommended to be presented in the manuscript.

3. Have the authors compare the results of the proposed method and the method is ref [9]?

Reviewer 2 Report

The paper developed a methodology for the location of charging stations and points for Electric trucks in the EU. This work was motivated by the need for transport mobility to cut down CO2 emissions. The study investigated three scenarios to predict the deployment of charging stations based on predicted traffic. The results could assist in planning and designing future charging infrastructure for electric road freight transport.

 The paper is interesting, and the technical content is sound.

 PRESENTATION COMMENTS:

Abstract

The abstract does not clearly tell the reader the approach used to solve the problem. Did not indicate any limitations

Introduction

1.         The introduction does not indicate the comparators used to demonstrate relevance.

  2.         The authors should consider listing the paper's contributions towards the end of the introduction.

 3.         Literature review needs to be increased with more details on the highlighted studies given.

The introduction does not state the unsolved problems and/or areas requiring improvement.

The introduction does not clearly state what has been done that has not been done before. 

The introduction does not indicate the comparators used to demonstrate relevance.

 The authors need to improve the clarity of presentation. Some statements are not clear. For example the statement on line 117 of page 4, line 186, line 201.

 Conclusions

4. There is no conclusions section. However, some conclusions are presented in the discussions section. Would it be better to have a concluding section?

Moreover, the discussions section is very brief. More insights should be given by the authors given the importance and the possible impact of the work.

 5. Any reservations and/or limitations have not been indicated.

TECHNICAL COMMENTS:

6. More details may be needed because even with the right tools, duplicating the work will not be possible.

 We request the authors to reconsider the statement on line 77 of page 2:

7. Some data may be available on some websites and databases. For example, https://ec.europa.eu/eurostat/web/transport/data/database and https://github.com/graphhopper/open-traffic-collection. Also, traffic for most cities is available from government databases. However, we agree that collecting such traffic data may be intractable.

 I believe that the work is very interesting. However, the authors need to add more discussion and presentation of the work in a more accessible manner. This will increase the impact of the work, which is topical. 

Additional comments on the highlighted document.

Reviewer 3 Report

In this paper, a public fast charging network for battery-electric trucks (BET) is designed and analyzed, The network design includes a node-based placement of charging locations and their scaling in terms of the number of required charge points. The analysis part uses three different scenarios for estimating the number of charge locations and charge points per location. In my view the work presented is straight forward and consistent. I have only few minor comments:

  • Lines 105-109: I could not quite follow the argumentation: 25% public charging leads to 50% public infrastructure use of BET. Please elaborate a bit more on this
  • Related to equation (1): clearly, the target d_CL is 50 km and 100 km, respectively. It would be interesting to see the actual distances (model output) between the charging locations, though.

Best regards.

Round 2

Reviewer 2 Report

The authors have addressed my concerns. Thanks for improving the work.